# Evaluation of the Effectiveness of Mesenchymal Stem Cells of the Placenta and Their Conditioned Medium in Local Radiation Injuries

**DOI:** 10.3390/cells9122558

**Published:** 2020-11-29

**Authors:** Vitaliy Brunchukov, Tatiana Astrelina, Daria Usupzhanova, Anna Rastorgueva, Irina Kobzeva, Victoria Nikitina, Sergei Lishchuk, Elena Dubova, Konstantin Pavlov, Valentin Brumberg, Marc Benderitter, Alexander Samoylov

**Affiliations:** 1State Research Center Burnasyan Federal Medical Biophysical Center of Federal Medical Biological Agency, 123098 Moscow, Russia; brunya2008@yandex.ru (V.B.); usupzhanova94@mail.ru (D.U.); rastorgueva.ann@gmail.com (A.R.); irina-kobzeva@yandex.ru (I.K.); nikitinava@yandex.ru (V.N.); leycom@mail.ru (S.L.); dubovaea@gmail.com (E.D.); kpavlov@fmbcfmba.ru (K.P.); brumb1225@gmail.com (V.B.); asamoilov@fmbcfmba.ru (A.S.); 2Institute for Radiological Protection and Nuclear Safety (IRSN), 92260 Fontenay-aux-Roses, France; marc.benderitter@irsn.fr

**Keywords:** mesenchymal stem cells, local radiation injuries, conditioned medium, cell technologies, X-ray radiation, skin, placenta

## Abstract

Background: The search for an effective therapy for local radiation injuries (LRI) is urgent; one option is mesenchymal stem cells (MSC) derived from the placenta and their conditioned medium for the regenerative processes of the skin. Methods: We used 80 animals, randomly assigned to four groups: control (C) animals that did not receive therapy; control with the introduction of culture medium concentrate (CM); introduction of MSCs (PL); introduction of CMPL. LRI modeling was performed on an X-ray machine at a dose of 110 Gy. Histological and immunohistochemical tests were performed. Results: On the 112th day, the area of the open wound surface in the CMPL group was 6.7 times less than in the control group. Complete healing of the open wound surface of the skin in the CM group was observed in 40%, in CMPL 60%, in the PL group 20%, and in the C group there were no animals with a prolonged wound defect. A decrease in inflammatory processes was observed in the CMPL group. Conclusions: the use of a concentrate of conditioned MSCs (CMPL group) in severe LRI in laboratory animals accelerates the transition of the wound process to the stage of regeneration and epithelization.

## 1. Introduction

Today, ionizing radiation sources are widely used in various fields of human activity, and their scope is constantly expanding, which increases the risk of radiation damage [1,2]. Studies have shown that radiation damage to the skin causes damage to the stem and proliferating cells of the epidermis, as well as in the vessels of the microcirculatory bed [3,4,5], so the final effect of ionizing radiation is determined by the balance between damage to the cells and recovery processes in the affected area and adjacent tissues [4,6].

Radiation to human skin in doses exceeding 8 Gy may lead to the development of local radiation injuries (LRI) [4]. In radiotherapy of oncological diseases, LRI is registered in 20–40% of cases [7]. Skin LRIs are characterized by the development of recurrent ulcers with pain syndrome, which significantly lengthens the treatment process due to persistent damage to blood and lymphatic vessels with the progression of tissue fibrosis, which worsens the results of treatment and the quality-of-life of patients [7]. Currently, there are no effective treatments for LRI.

Taking into account the pathogenetic mechanisms of radiation-induced lesions, the use of cellular technologies using mesenchymal stem cells (MSCs) and their waste products (paracrine factors) may become a promising method of treating skin LRI [5]. MSCs are capable of self-renewal and various types of differentiation in the adipogenic, osteogenic, chondrogenic, and myogenic directions [1]. The use of MSCs leads to the healing of the wound surface of the skin and its appendages, diabetic ulcers, damage to skeletal muscles and cartilage, and the heart. Intravenous, local administration of MSCs helps to reduce necrotic changes, reduce inflammation, and significantly improve the processes of granulation, reepithelialization, neoangiogenesis, and hair restoration [1,3,4].

The main effect of MSCs may be due to their secretory activity, associated with the production of a wide range of cytokines and growth and angiogenic factors [8]. Paracrine factors initiate the stimulation of host MSCs, triggering the regeneration of damaged tissues. Thus, the cytokines involved in the regulation of the inflammatory process include IL-1β, 4, 6, 10, 12, and 17, TNF-α, TGF-β1, PGE-2, PDGF, HGF and SDF-1. VEGF, FGF-2, EGF, TGF-α, HGF, IGF-1, etc., are responsible for neoangiogenesis, and the regulation of tissue fibrosis involves the participation of IL-4, 16, TGF-β1, HGF, bFGF, etc. MSC secretome injections in the form of conditioned media containing extracellular vesicles also have positive effects, as well as the MSCs themselves. Further assessment of the paracrine potential of MSCs may open up new ways of treating acute and chronic forms of MLP of the skin [1,3,4].

One of the most important advantages of MSCs is their low immunogenicity, which allows the use of allogeneic MSCs without the risk of rejection reactions. Sources of MSCs are various human tissues (bone marrow, adipose tissue, skin, placenta, synovial membrane, cartilage, etc.) [1,3,4,6,7]. The main sources of MSCs are bone marrow, mucosal and placental tissues, etc.

Placental tissue is of great interest due to the simplicity of sampling, the absence of ethical problems, and the ability to quickly obtain and accumulate the necessary amount of cellular material. MSCs derived from the placenta are known to have a higher regenerative potential compared to cells from other sources, but there are no data on the use of these cells in LRI [9].

Thus, the presented data indicate that MSCs derived from the placenta and the paracrine factors produced by them can be used to produce drugs intended for the treatment of LRI, which undoubtedly deserves further study.

The aim of this study was to study the effect of human MSCs derived from the placenta and their conditioned medium concentrate on skin regenerative processes in laboratory animals with LRI.

## 2. Materials and Methods

### 2.1. Study Groups

There were 80 laboratory animals used in the study (Wistar male rats aged 8–12 weeks and weighing 210.0 ± 30.0 g). The animals were obtained from the specialized laboratory animal nursery Pushchino, had the appropriate veterinary certificate, and were quarantined for 14 days. The study was approved by the section of the Academic Council (extract No. 43A, dated 25/9/2017) and at the meeting of the local bioethical committee (Protocol No. 8b dated 10/11/2012) of the State Research Center—Burnasyan Federal Medical Biophysical Center of Federal Medical Biological Agency.

Laboratory animals were randomized and divided into four groups (20 animals each) depending on the type of therapy performed:

Group 1—control (C)-irradiated rats without subsequent therapy;

Group 2—control (CM)-irradiated rats that received intradermal administration of culture medium (MesenCult) concentrate around the affected area three times, on days 1, 14, and 21;

Group 3—irradiated rats that received intradermal administration of human MSCs derived from the placenta (PL) three times, on days 1, 14, and 21;

Group 4—irradiated rats that received intradermal administration of concentrate of conditioned medium (CMPL) MSCs derived from the placenta three times on days 1, 14, and 21.

Each laboratory animal was observed 17 times: 1, 7, 14, 21, 28, 35, 42, 49, 56, 63, 70, 77, 84, 91, 98, 105, and 112 days after LRI modeling. During the examination, the condition of the laboratory animal was monitored with an assessment of its behavior, movement, cardiovascular and/or respiratory functions, changes in appetite and weight, body temperature, etc. On set days, the skin surface was examined and the course of the wound process was evaluated (the depth of damage to the skin, their size (length, width), the total area of the changed skin, the area of the open wound surface, the presence of discharge, blisters, scab, exfoliated epidermis, the color of the exposed dermis, fibrin plaque).

Animals were removed from the experiment 28, 42, 56, 70, 91, and 112 days from the beginning of the experiment.

### 2.2. Modeling of LRI

Modeling of relatively “soft” X-ray radiation of LRI was carried out on an LNK-268 (RAP100-10) X-ray unit (Diagnostika-M LLC, Moscow, Russia) with a radiation exposure mode at a dose of 110 Gy with a 0.1 mm aluminum filter, voltage 30 kV, beam current 6.1 mA, dose rate 21.1 Gy/min for 312 s (dose accuracy ± 5%, dose measurement uncertainty ± 6%) according to the proposed earlier method [10,11,12], leading to a short latency period and chronic skin ulcer in laboratory animals. After irradiation, the animals were seated in individual sterile boxes with an autonomous Smart Flow ventilation system (Tecniplast Group, Buguggiate, Italy), with free access to water and food.

### 2.3. Cultivation of MSCs

The experiment used nonpersonalized human MSCs derived from the placenta samples that are under long-term cryopreservation in a Cryobank. MSCs were cultured in a medium without xenogenic components (Stem Cell, Vancouver, Canada) with the addition of 100 U/mL of penicillin, 100 U/mL of streptomycin, and 2 mm of glutamine from the 3rd to the 5th passage. The resulting MSCs were administered to laboratory animals at a calculated dose of 2 million cells per 1 kg.

### 2.4. Immunological Characteristics and Viability of MSCs

The MSCs derived from the placenta immunophenotype was determined by flow cytometry. The expression of surface markers was evaluated using fluorochrome-labeled antibodies against CD34, CD45, CD90, CD105, CD73, and HLA-DR (BD Biosciences and Becman Coulter, Brea, CA, USA) on a FACSCanto II flow cytometer (Becton Dickinson, Durham, CA, USA) in accordance with the manufacturer’s instructions.

Cell viability was evaluated using a 7-ADD dye that penetrates the cell’s cytoplasmic membrane and binds to its DNA. The number of CD45-negative/7-ADD-positive cells was determined on a FACS Canto II flow cytometer (Becton Dickinson) in accordance with the manufacturer’s instructions.

### 2.5. Obtaining a Conditioned Medium

The conditioned medium of MSCs derived from the placenta (CMPL) was taken into sterile tubes at 3–5 passages when the cells reached 80–90% confluence. A laboratory tangential flow filtration system, LabScale, developed for concentration, diafiltration, and microfiltration, was used to obtain CMPL. The CMPL was placed in a tangential flow filtration system and concentrated 8.08 times with an inlet pressure of 40–52 psi and an outlet pressure of 8–12 psi. The resulting volume was passed through a nylon syringe filter with a pore size of 0.22 μm (Corning, New York, NY, USA). The protein concentration was 648 μg/mL, IL-6: 853 pg/mL, IL-8: 8730 pg/mL, IL-10: 17.7 pg/mL, TGF-β: 1.0 pg/mL. The volume concentrate of conditioned medium for rats of the CMPL group for each injection was 0.4 mL. The introduction was carried out intradermally up to 12 points around the irradiation zone, retreating 2–3 mm from the edge.

### 2.6. Histological and Immunohistochemical Study

A skin flap was excised from the affected area (the area of the wound defect, with the adjacent skin and underlying muscles) and fixed in a 10% solution of neutral formalin. Further processing of excised samples was performed using standard histological methods. Preparations stained with hematoxylin and eosin were used for general assessment of the condition of the studied tissues.

Immunohistochemical examination of tissue samples on days 28 and 112 of the experiment was performed using an automated method, a Ventana BenchMark Ultra immunostainer with dewaxing and unmasking in an apparatus using antibodies to VEGF (Novocastra, Milton Keynes, UK, skin blood vessel endothelium marker), CD31 (Novocastra, endothelial cell marker), CD68 (Novocastra, macrophage marker), PGP9.5 (Novocastra, marker of differentiating neurons in the skin), Ki67 (Novocastra, marker of cell proliferation), FVIII (Novocastra, marker for platelet adhesion factor), collagens of types I and III (Novocastra), tissue inhibitor of metalloproteinases TIMP2 (Novocastra), and metalloproteinases of types 2 and 9 (Novocastra). Marker expression was evaluated semiquantitatively, assigning scores from 0 to 3, where 0 means no expression and 3 means fully expressed.

Statistical analysis of the results was performed using Microsoft Office Excel 2007 (Redmond, WA, USA), Statistica 6 (Round Rock, TX, USA) and ImageTool software (San Antonio, TX, USA. The Mann‒Whitney test was used to assess the significance of the differences. 

## 3. Results

### 3.1. Immunological Characteristics and Viability of MSCs

When analyzing the MSCs immunophenotype using flow cytometry, a high expression of MSCs markers (CD73, CD90, CD105) was detected in all cell cultures; markers of hematopoietic and lymphocytic origin were absent (CD34, CD45, HLA-DR). The immunophenotype met the requirements of the International organization of cell therapy for human MSCs [1]. MSCs maintained high activity and viability (98.21 ± 1.72%, 7-ADD) throughout the entire culture period (Figure 1).

### 3.2. Planimetric Analysis

The majority of animals in the study groups on day 7 showed a clearly visible area of altered skin, outlined by a demarcation line, with signs of dry or wet dermatitis. The average area of the changed skin was significantly lower in the C and CM groups compared to the PL and CMPL groups (*p* ≤ 0.05) (Figure 2a). However, by day 14, the total changed skin area in the C, PL, and CMPL groups did not differ, whereas in the CM group it was less until the end of the experiment (Table 1).

From the 14th day of the experiment, the open wound surface of the skin was recorded in all groups of animals. The dynamics of reducing the area of the open wound surface was the same for all groups up to day 42 of the study. After that, we observed wave dynamics of increase and decrease of the open wound skin surface in all groups except CM (Figure 2b, Table 2).

On day 112, the area of the open wound surface in the CMPL group was 6.7 times smaller than in the control group. Complete healing of the open wound surface of the skin in the CM groups was observed in 40% and the CMPL in 60%, in the PL group in 20%, and in the C group there were no animals with a prolonged wound defect (Figure 2c).

### 3.3. Histological Examination

In all groups, an open wound defect covered with a purulent‒necrotic crust was formed on the 28th day after irradiation. Weak, mainly perivascular lymphocytic‒plasmocytic infiltration with an admixture of single neutrophilic granulocytes and moderate vascular proliferation of the microcirculatory bed was detected in the underlying dermis in the area of the defect bottom. Moderate thickening of the adjacent epidermis was noted along the edges of the wound defect, and few intraepidermal lymphocytes were recorded. At the same time, only in the CMPL group was the “creeping” of regenerating epithelium from one of the wound edges in the form of a strip 3–4 epithelial cells thick noted (Figure 3).

In group C, on day 56, the bottom of the skin defect reached the large subcutaneous muscle, and, in some cases, the subcutaneous fat. Pronounced edema and lymphocytic‒plasmocytic infiltration of muscle tissue were determined. In the connective tissue of the dermis in the area of the edges of the wound defect, lymph‒histiocytic infiltration with an admixture of neutrophils, granulation, and proliferation of microvessels were detected. The epidermis adjacent to the wound defect was thickened to 10–12 layers of cells, and focal hyperkeratosis, acanthosis, and degenerative changes in keratinocytes were noted. Areas of epithelial regeneration with a thickness of 4–8 epithelial cells were determined in the area of the defect edges (Figure 3(1b)); by the 112th day, pronounced purulent‒necrotic changes in soft tissues appeared in the area of the defect bottom. Underlying connective tissue and moderate lymphocytic‒plasmocytic infiltration were observed, with an admixture of neutrophilic granulocytes, moderate vascular proliferation of the microcirculatory bed, focal edema, and pronounced fibrotic changes. In most cases, areas of fibrosis and weakly expressed lymphoplasmocytic infiltration in the area of the large subcutaneous muscle and Hypoderma were detected. At the edges of the wound defect, there were large areas of regeneration of the integumentary epithelium in the form of a layer of cells 1–2 epithelial cells thick. The adjacent epidermis was thickened (up to 6–11 layers of cells), with signs of vacuole dystrophy and acanthosis (Figure 3(1c)).

In the CM group, signs of epithelization along the edges of the defect were noted on day 56 in most samples; in some cases a deep skin defect remained, which reached the subcutaneous fat and was covered with a purulent‒necrotic crust. In all preparations, the large subcutaneous muscle was of the usual histological structure with moderate edema, subcutaneous fat with moderate edema, or moderate lymph with plasmocytic infiltration. In the underlying dermis, there were areas of fibrosis and weak perivascular lymphocytic infiltration (Figure 3(2b)). By the 112th day, only one observation revealed a large skin defect covered with a purulent‒necrotic crust, which reached the subcutaneous fat with necrosis in the superficial parts and clusters of hemosiderophages in the deeper parts. In other preparations, the skin defect was partially or completely epithelized. In the dermis, there were areas of fibrosis and mild perivascular lymphocytic infiltration. In all cases, the adjacent epidermis was thickened (up to 10–11 layers of cells), with signs of severe dystrophy (Figure 3(2c)).

In the PL group on day 56, the bottom of the skin defect showed necrotic dermis, striated muscle, and underlying adipose tissue with pronounced neutrophilic infiltration. At the edges of the defect, there was moderate lymph‒plasmocytic infiltration with an admixture of neutrophilic granulocytes, moderate proliferation of microcirculatory vessels, granulation, and fibrosis of striated muscle tissue (Figure 3(3b)). By the 112th day, the extensive skin defect was covered with a purulent‒necrotic crust in all cases. Its bottom is represented by fibrotic connective tissue with angiomatosis, granulations, moderate lymph‒plasmocytic infiltration with an admixture of neutrophilic granulocytes, and microvessel proliferation. The striated tissue at the bottom of the defect was not detected. There were extensive areas of fibrosis of the underlying adipose tissue. The epidermis at the edge of the wound defect was thickened (up to 8–10 layers of cells), with signs of vacuole dystrophy and proliferation of hair follicles (Figure 3(3c)).

In the CMPL group, on day 56, the open wound skin defect was covered with a purulent‒necrotic crust, marginal epithelization was recorded over a longer length in most samples, and the thickness of the epithelial layer was 5–8 cells. The underlying dermis was moderately fibrotic with focal subepithelial edema and the presence of hair follicle rudiments in the amount of 1–3 in the field of vision. In the area of the defect bottom, muscle and adipose tissue were completely replaced by fibrous tissue with granulations with moderate lymph‒plasmocytic infiltration with an admixture of neutrophilic granulocytes and pronounced microvessel proliferation (Figure 3(4b)). By the 112th day in all samples, the skin defect was completely epithelized and the thickness of the epithelial layer was 5–7 cells. The underlying dermis was focally fibrotic. There were rudiments of hair follicles (1–3) in the field of vision with focal proliferation of microcirculatory vessels. Large subcutaneous muscle was not detected in the central parts; it was replaced by connective tissue. There were no inflammatory changes (Figure 3(4b)).

### 3.4. Immunohistochemical Study

As a result of an immunohistochemical study, it was found that the number of newly formed vessels in whose endothelial cells the expression of CD31 was determined increased from day 28 to day 112 in the PL and CMPL groups (from 2.6 ± 1.0 to 10.97 ± 1.6 and from 4.1 ± 0.6 to 8.2 ± 1.8, respectively, *p* ≤ 0.05), which indicated an increase in neoangiogenesis by the end of the experiment. Such changes were not detected in groups C and CM (Figure 4), nor for the vascular endothelial growth factor (VEGF) in endothelial cells and stroma cells in all the studied groups, except for CMPL (Figure 4d and Figure 5b). In groups C, CM, and PL, an increase in FVIII expression in vascular endothelial cells was observed by day 112 of the experiment (*p* ≤ 0.05) (Figure 4b and Figure 5a).

In the course of the experiment, we noted an increase in the number of CD68-positive macrophages in the tissues surrounding the wound defect in groups C and PL (from 11.7 ± 1.4 and 12.9 ± 3.6 at 28 days to 24.73 ± 2.4 and 29.3 ± 3.5 at 112 days, respectively, *p* ≤ 0.05), while in the CM group it was determined by the decrease in the number of these cells (22.1 ± 1.6 and 13.07 ± 1.8, *p* ≤ 0.05), and in the CMPL group their number did not change (Figure 4a and Figure 5a).

The number of regenerating nerve fibers expressing PGP9.5 increased by the end of the experiment in the C, CM, and PL groups (*p* ≤ 0.05), and remained unchanged in the CMPL group (Figure 4a and Figure 5a).

Expression of matrix metalloproteinases (MMP) 2 and 9, which led to the destruction of extracellular matrix proteins and stimulated cell migration and reproduction, decreased in all groups at the end of the experiment, with the exception of the CMPL group (Figure 4b,d), while expression of TIMP2, which is a tissue inhibitor of MMP, increased by day 112 in the C and CM groups, decreased in the PL group, and remained unchanged in the CMPL group (Figure 4b,c). The expression of “mature” type I collagen in the stroma increased in all groups from day 28 to day 112, with the exception of the PL group, while the expression of “immature” type III collagen in the stroma decreased by day 112 in the C, CM, and PL groups, but did not change in the CMPL group (Figure 4b,c).

## 4. Discussion

Skin LRIs pose a serious medical, social, and economic problem. It is known that early effects of skin damage from ionizing radiation (dry and wet dermatitis) are associated with damage to the epidermis, and late effects (skin atrophy, radiation necrosis, etc.) are the result of damage to the dermis [4]. Thus, the radiation target of the epidermis is the highly radiosensitive cells of the basal layer, and the vessels of the microcirculatory bed in the dermis. As a result, with deep radiation burns, necrotic and degenerative processes cover all layers of the skin, gradually spreading to the underlying tissues, up to the bone.

This study is aimed at preventing complications of late radiation injuries, improving the nature of the treatment course, and speeding up the healing time of skin LRIs. The introduction of MSCs, culture, and conditioned media was performed three times: in the absence of changes in the skin (day 1), after the formation of radiation dermatitis (day 14), and after the appearance of the wound surface (day 21) to activate reparative processes and angiogenesis in damaged skin, reduce the healing time of ulcerative defects, and prevent relapses.

In the course of the present study, the use of a conditioned medium concentrate and culture medium showed greater effectiveness in terms of reducing the time and area of the open wound surface. The greatest number of cases of complete healing of the affected skin of animals by the end of the observation period (112 days) was detected in the CMPL group, and healing was also noted in the CM group. In the same groups, no clinical signs of LRI recurrence were detected, but not in the C and PL groups (Figure 3).

In our previous study [7], the effectiveness of the use of MSCs of the gingival mucosa in LRI was shown. The absence of a significant effect when using MSCs derived from the placenta in this study is due to the tissue specificity of this cell source and the peculiarities of their production of paracrine factors.

According to the histological study, there was a decrease in inflammatory processes, the presence of hair follicle rudiments, and proliferation of microcirculatory vessels in the CMPL group, in contrast to other groups in which these changes were not so noticeable.

Given the nature of the radiation damage in all groups, the regenerative potential of cells in the affected area is significantly reduced, which was confirmed by data from immunohistochemical research. An increase in the expression of metalloproteinases (MMP 2 and 9), TIMP 2, collagen I and III, as well as the number of CD68 macrophages, was observed in the CMPL group on day 112, which probably indicates an increase in the rate of scarring and healing of the wound surface, which is confirmed by planimetric studies (Figure 4a,b).

## 5. Conclusions

Thus, the use of MSCs derived from the placenta conditioned medium concentrate (CMPL group) in severe LRI in laboratory animals accelerates the transition of the wound process to the stage of regeneration and epithelization. Interestingly, in one of the control groups, when using a culture medium concentrate (CM group), a significant decrease in the area of the wound surface was observed compared to other groups during the entire observation period. However, the analysis of histological and immunohistochemical studies does not allow us to unequivocally assert the effectiveness of this type of therapy.

## Figures and Tables

**Figure 1 cells-09-02558-f001:**
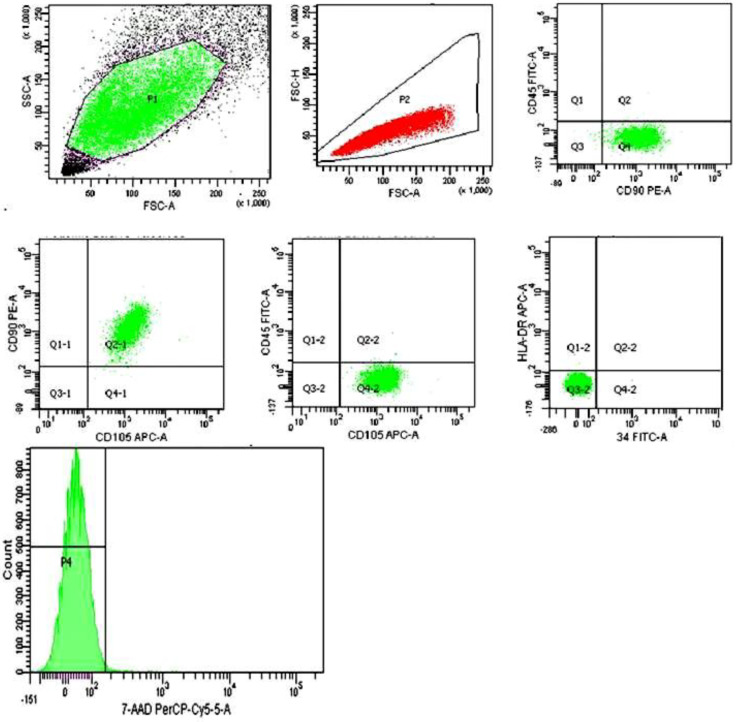
MSCs derived from the placenta immunophenotype: CD90+/CD105+73+/CD45−/CD34−/HLA-DR−, 7−ADD (99.5%).

**Figure 2 cells-09-02558-f002:**
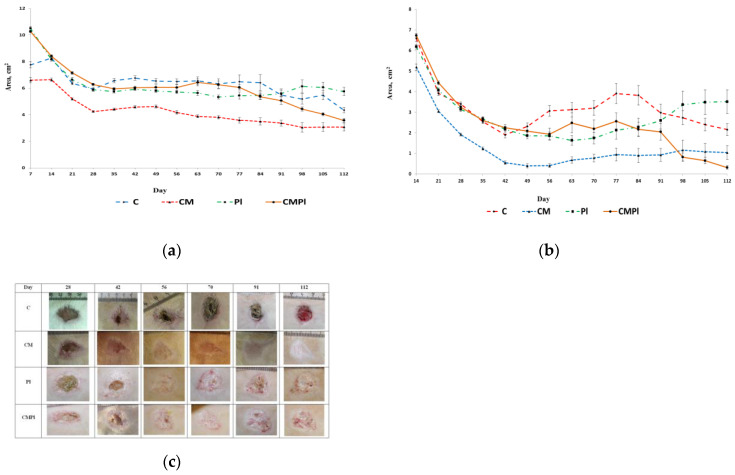
Dynamics of the LRI of animals: (**a**) dynamics of development of visible region of rat skin changes that occurred under X-ray radiation; (**b**) dynamics of development of an open wound surface of rat skin after exposure to X-ray radiation; (**c**) dynamics of healing of an open wound surface of the skin in animals (C: irradiated rats without subsequent therapy, CM: irradiated rats that received intradermal administration of culture medium (MesenCult) concentrate, PL: therapy of LRI using MSC derived from the placenta, CMPL: therapy of LRI using a concentrate of conditioned medium collected from a culture of the MSCs).

**Figure 3 cells-09-02558-f003:**
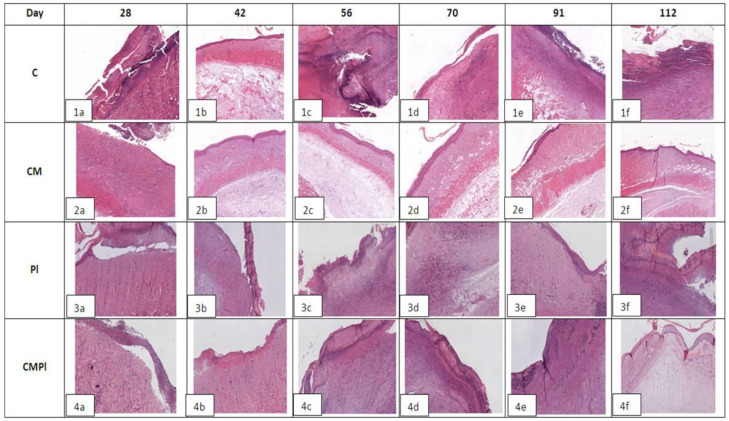
Histological examination of animal skin defects. Hemotoxylin‒eosin staining, ×40 magnification for C and CM groups, ×20 for PL and CMPl groups (C: irradiated rats without subsequent therapy, CM: irradiated rats that received intradermal administration of culture medium (MesenCult) concentrate, PL: therapy of LRI using MSCs derived from the placenta, CMPL: therapy of LRI using a concentrate of conditioned medium collected from the culture of the MSCs).

**Figure 4 cells-09-02558-f004:**
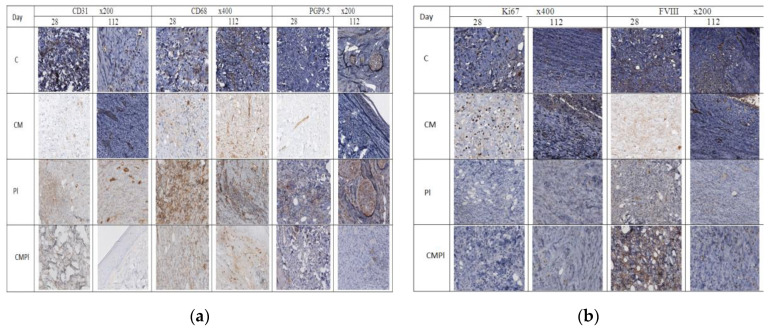
Immunohistochemical study of animal skin defects: (**a**) CD31, CD68, PGP9.5; (**b**) Ki67 and FVIII; (**c**) TIMP2, Collagen I, Collagen III, and FVIII; (**d**) VEGF, MMP2, and MMP9. (C: irradiated rats without subsequent therapy, CM: irradiated rats that received intradermal administration of culture medium (MesenCult) concentrate, PL: therapy of LRI using MSC derived from the placenta, CMPL: therapy of LRI using a concentrate of conditioned medium collected from the culture of the MSCs).

**Figure 5 cells-09-02558-f005:**
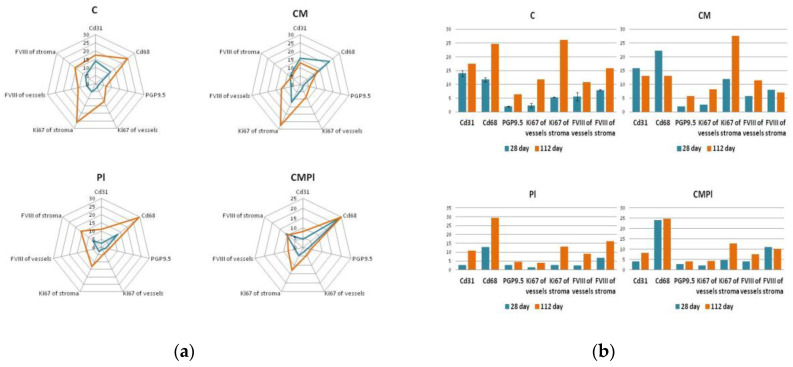
Immunohistochemical study of animal skin defects: (**a**) Absolute number of IHC markers in excised animal skin defect tissue samples per 10 visual fields; (**b**) semiquantitative assessment of the expression of IHC markers in excised animal skin defect samples (with a score from 0 to 3, where 0 is the absence of expression and 3 is full expression). (C: irradiated rats without subsequent therapy, CM: irradiated rats that received intradermal administration of culture medium (MesenCult) concentrate, PL: therapy of LRI using MSC derived from the placenta, CMPL: therapy of LRI using a concentrate of conditioned medium collected from culture of the MSCs. Blue: 28 days; orange: 112 days from the start of therapy. The score is from 0 to 3, where 0 is the absence of expression and 3 is full expression).

**Table 1 cells-09-02558-t001:** The area of the total changed surface of animal skin in LRI (cm^2^).

Day	C	CM	Pl	CMPL
7	7.76 ± 0.47	6.60 ± 0.42	10.5 ± 0.31 ^1,2^	10.27 ± 0.19 ^1,2^
14	8.26 ± 0.33	6.63 ± 0.26	8.16 ± 0.25	8.41 ± 0.21 ^2^
21	6.36 ± 0.23	5.19 ± 0.18 ^1^	6.63 ± 0.36 ^2^	7.15 ± 0.2 ^1,2^
28	5.93 ± 0.27	4.25 ± 0.13 ^1^	5.88 ± 0.22 ^2^	6.28 ± 0.18 ^2^
35	6.56 ± 033	4.42 ± 0.0.17	5.74 ± 0.21 ^2^	5.95 ± 0.19 ^2^
42	6.76 ± 0.34	4.59 ± 0.21	5.92 ± 0.19	6.04 ± 0.21
49	6.54 ± 0.35	4.62 ± 0.24	5.80 ± 0.21	6.06 ± 0.39
56	6.50 ± 0.41	4.18 ± 0.26 ^1^	5.71 ± 0.24 ^2^	6.05 ± 0.41 ^2^
63	6.56 ± 0.60	3.88 ± 0.23	5.64 ± 0.37 ^2^	6.43 ± 0.54 ^2^
70	6.33 ± 0.72	3.81 ± 0.26 ^1^	5.33 ± 0.29	6.26 ± 0.58
77	6.49 ± 1.04	3.59 ± 0.43	5.44 ± 0.53	6.06 ± 0.41 ^2^
84	6.41 ± 1.21	3.50 ± 0.54	5.45 ± 0.58	5.36 ± 0.47 ^2^
91	5.50 ± 0.88	3.38 ± 0.47	5.60 ± 0.52 ^2^	5.07 ± 0.58 ^2^
98	5.19 ± 0.79	3.05 ± 0.71	6.13 ± 0.95	4.43 ± 0.31
105	5.47 ± 0.69	3.08 ± 0.67	6.06 ± 0.81	4.04 ± 0.23
112	4.35 ± 0.42	3.07 ± 0.60	5.75 ± 0.61	3.57 ± 0.30

Notes: ^1^ Significant differences in all groups compared to the control (C) (*p* ≤ 0.05). ^2^ Significant differences between the PL and CMPL groups compared to the CM group (*p* ≤ 0.05).

**Table 2 cells-09-02558-t002:** Area of open wound surface of animal skin in LRI (cm^2^).

Day	C	CM	Pl	CMPL
14	6.59 ± 0.28	5.19 ± 0.33 ^1^	6.19 ± 0.23 ^2^	6.73 ± 0.18 ^2^
21	3.91 ± 0.2	3.05 ± 0.14 ^1^	4.07 ± 0.22 ^2^	4.42 ± 0.19 ^1,2^
28	3.40 ± 0.19	1.92 ± 0.14 ^1^	3.14 ± 0.2 ^2^	3.25 ± 0.18 ^2^
35	2.54 ± 0.19	1.23 ± 0.15 ^1^	2.68 ± 0.21 ^2^	2.60 ± 0.21 ^2^
42	1.90 ± 0.28	0.55 ± 0.16 ^1^	2.17 ± 0.27 ^2^	2.25 ± 0.27 ^2^
49	2.32 ± 0.34	0.39 ± 0.19 ^1^	1.86 ± 0.29 ^2^	2.09 ± 0.42 ^2^
56	3.07 ± 0.53	0.41 ± 0.17 ^1^	1.85 ± 0.37 ^2^	1.94 ± 0.59 ^2^
63	3.12 ± 0.70	0.68 ± 0.32 ^1^	1.63 ± 0.46	2.48 ± 0.96
70	3.20 ± 0.71	0.78 ± 0.37 ^1^	1.74 ± 0.55	2.19 ± 0.86
77	3.91 ± 0.97	0.94 ± 0.62 ^1^	2.12 ± 0.86	2.56 ± 0.80
84	3.82 ± 0.95	0.90 ± 0.69 ^1^	2.28 ± 0.87	2.17 ± 0.69
91	2.98 ± 0.83	0.93 ± 0.62 ^1^	2.60 ± 0.81	2.05 ± 0.80
98	2.74 ± 0.69	1.16 ± 0.97 ^1^	3.37 ± 1.32	0.82 ± 0.40 ^1,2^
105	2.40 ± 0.62	1.09 ± 0.79 ^1^	3.49 ± 1.21 ^2^	0.65 ± 0.32 ^1,2,3^
112	2.15 ± 0.57	1.04 ± 0.68	3.52 ± 1.14 ^2^	0.32 ± 0.18 ^1,2,3^

Notes: ^1^ Significant differences in all groups compared to the control (C) (*p* ≤ 0.05). ^2^ Significant differences between the PL and CMPL groups compared to the CM group (*p* ≤ 0.05). ^3^ Significant differences between the PL and CMPL groups (*p* ≤ 0.05).

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
