# Peer review of "Evaluation of the Effectiveness of Mesenchymal Stem Cells of the Placenta and Their Conditioned Medium in Local Radiation Injuries"

_cells, 2020, doi:10.3390/cells9122558_

Round 1
Reviewer 1 Report
While I welcome any research on the regenerative potential of MSCs, I have massive criticisms and major concerns of this study:
The whole study must be reviewed by a native speaker. Beginning with the first sentence inthe Abstract, there are many misunderstood and poorly expressed sentences.
Revise the whole abstract. describe all methods in an appropriate manner.
Methods:
Describe all methods in an appropriate manner, e.g. line 97....a medium ....which one, please specify.
Please describe the conditioned medium and especially the production of the CM before administration (lines 115-118 do not adequately describe this, e.g. culture time, volumes, ......)
line100: Did you really administered 2Mio cells per kg ....that would be around 20,000 cells per intradermal administration (with an average of 20g live weight per mouse). PÜlease check and describe in an appropriate manner.
You should also state that the cells were administered intradermally...please describe this procedure.
Methods: Please describe characterication of the MSCs in an appropriate manner, light microscopy, charcteristic histograms of the flow cytometric results.
Statistical analysis is not described. Which methods are used?
Fig. 1 and 3: Please enhance quality of the figure
Fig. 3: Maybe you should split the figure in more figures. the current presentation is not clear.
Figs: Please add bars.
Minor:
Authors: which authors contribute equally? Not stated!
Abstract: please describe the animals used in the abstract
line 68:.....randomized randomly....??
line 99: 2 mM , not 2mm.
line112: PQ?
Author Response
Response to Reviewer 1 Comments
Dear REVIEWER,
Thank you for your work on the article. We revised the manuscript corrections are highlighted and figures in the attached files.
With best regards,
Authors.
And we clarify following questions:
Point 1: Comments and Suggestions for Authors
While I welcome any research on the regenerative potential of MSCs, I have massive criticisms and major concerns of this study: The whole study must be reviewed by a native speaker. Beginning with the first sentence in the Abstract, there are many misunderstood and poorly expressed sentences.
Revise the whole abstract. describe all methods in an appropriate manner.
Methods: Describe all methods in an appropriate manner, e.g. line 97....a medium ....which one, please specify.
Response 1: We agree and have made some corrections:
Medium: MesenCult. Data supplemented.
Point 2: Please describe the conditioned medium and especially the production of the CM before administration (lines 115-118 do not adequately describe this, e.g. culture time, volumes, ......)
Response 2: We agree and have made some corrections:
Getting a conditioned medium
The conditioned medium of placental MSCs (CMPL) was taken into sterile tubes at 3-5 passages when the cells reached 80-90% confluence. A laboratory tangential flow filtration system LabScale, developed for concentration, diafiltration, and microfiltration, was used to obtain CMPL. The CM was placed in a tangential flow filtration system and concentrated 8.08 times with an inlet pressure of 40-52 psi and an outlet pressure of 8-12 psi. The resulting volume was passed through a nylon syringe filter with a pore size of 0.22 μm (Corning, USA). The protein concentration was 648 μg / ml, IL-6 - 853 pg / ml, IL-8 - 8730 pg / ml, IL-10 - 17.7 pg / ml, TGF-β - 1.0 pg / ml. The volume concentrate of conditioned medium for rats of the CMPL group for each injection was 0.4 ml. The introduction was carried out intradermally up to 12 points around the irradiation zone, retreat 2-3 mm from the edge.
Point 3: line100: Did you really administered 2Mio cells per kg ....that would be around 20,000 cells per intradermal administration (with an average of 20g live weight per mouse). Please check and describe in an appropriate manner.
Response 3: We agree and have made some corrections:
Cultivation of MSC
In the experiment, we used non-personalized samples of MSCs from the human placenta, which are on long-term cryostorage in a Cryobank. MSCs were cultivated in a medium without xenogenic components - MesenCult (Stem Cell, Canada) with the addition of 100 U / ml of penicillin 100 U / ml of streptomycin; 2 mM glutamine from the 3rd to the 5th passage. The dose of MSCs for injection in each rat was 0.4x106. A suspension of MSCs in 0.5 ml of a sterile 0.9% NaCl solution was injected intradermally up to 12 points around the irradiation zone, 2-3 mm apart from the edge.
Point 4: You should also state that the cells were administered intradermally...please describe this procedure.
Response 4: We agree and have made some corrections:
These data were supplemented in the paragraph «Obtaining MSCs» in the manuscript.
Point 5: Methods: Please describe characterication of the MSCs in an appropriate manner, light microscopy, charcteristic histograms of the flow cytometric results.
Response 5: There are many pictures in the article. Especially for you we can send a file to review our results. However, we decided not to insert this data so as not to overload the article.
Figure 1. Placental MSC immunophenotype: CD90 + / CD105 + / CD45− / CD34− / HLA-DR−, 7 − ADD (99.5%)
Point 6: Statistical analysis is not described. Which methods are used?
Response 6: We agree and have made some corrections: Statistical analysis of the results was carried out using Microsoft Office Excel 2007, Statistica 6, ImageTool software. The Mann-Whitney test was used to assess the significance of the differences.
Point 7: Fig. 1 and 3: Please enhance quality of the figure
Response 7: We agree and have made some corrections:
We will provide these figures in good quality in a separate file.
Point 8: Fig. 3: Maybe you should split the figure in more figures. the current presentation is not clear.
Response 8: We agree and have made some corrections:
We will provide these figures in good quality in a separate file.
Point 9: Figs: Please add bars.
Response 9: We agree and have made some corrections:
Figure: 1. a) Dynamics of development of visible region rats skin changes which was occurred under X-ray radiation, b) Dynamics of development of an open wound surface of the rats skin after exposure to X-ray radiation, c) Dynamics of healing of an open wound surface of the skin in animals (C - irradiated rats without subsequent therapy, CM - irradiated rats was received intradermal administration of culture medium (MesenCult) concentrate, PL - therapy of LRI using MSC of placenta, CMPL – therapy of LRI using a concentrate of conditioned medium collected from culture of the placental MSC)
Figure 3. Histological examination of animal skin defects. Hemotoxylin-eosin staining, X40 magnification for C and CM groups, X20 - for PL and CMPl groups (C - irradiated rats without subsequent therapy, CM - irradiated rats was received intradermal administration of culture medium (MesenCult) concentrate, PL - therapy of LRI using MSC of placenta, CMPL – therapy of LRI using a concentrate of conditioned medium collected from culture of the placental MSC)
Figure 4. Immunohistochemical study of animal skin defects: (a) Immunohistochemical study of CD31, CD68, PGP9.5 of animal skin defects; (b) Immunohistochemical study of Ki67 and FVIII of animal skin defects; (c) Immunohistochemical study of TIMP2, Collagen I, Collagen III and FVIII of animal skin defects; (d) Immunohistochemical study of VEGF, MMP2 and MMP9 of animal skin defects (C - irradiated rats without subsequent therapy, CM - irradiated rats was received intradermal administration of culture medium (MesenCult) concentrate, PL - therapy of LRI using MSC of placenta, CMPL – therapy of LRI using a concentrate of conditioned medium collected from culture of the placental MSC)
Figure 5. Immunohistochemical study of animal skin defects: (a) The absolute number of IHC markers in samples of excised tissues of a skin defect of animal skin per 10 fields of view; (b) Semi-quantitative assessment of the expression of IHC markers in excised samples of animal skin defect (C - irradiated rats without subsequent therapy, CM - Irradiated rats was received intradermal administration of culture medium (MesenCult) concentrate, PL - therapy of LRI using MSC of placenta, CMPL – therapy of LRI using a concentrate of conditioned medium collected from culture of the placental MSC. Blue - 28 days; orange - 112 days from the start of therapy. In score from 0 to 3, where 0 is the absence of expression, and 3 is the expressed expression).
Point 10: Minor:
Authors: which authors contribute equally? Not stated!
Response 10: We have noted the contribution of each author to this study before the abstract, as well as in the section "Author Contributions"
Point 11: Abstract: please describe the animals used in the abstract
Response 11: The requirements for annotation formatting are very strict, we cannot go beyond a certain number of characters. Therefore, an extended description of groups with rats is indicated in the materials and methods:
Laboratory animals were randomized and divided into 4 groups (20 animals each) depending on the type of therapy performed:
Group 1 - control (C), in which the irradiated animals did not receive therapy;
Group 2 - control. Irradiated rats with intradermal injection concentrate of MesenCult culture medium(CM) around the affected area three times on days 1, 14 and 21;
Group 3 - irradiated rats with intradermal administration of human placenta MSCs (PL) three times on days 1, 14, and 21;
Group 4 - irradiated rats with intradermal administration concentrate of conditioned medium (CMPL) MSCs of human placenta three times on days 1, 14, and 21.
Point 12: line 68:.....randomized randomly....??
Response 12: We agree and have made some corrections: All mistakes was fixed. Laboratory animals were randomized and divided into 4 groups (20 animals each) depending on the type of therapy performed.
Point 13: line 99: 2 mM , not 2mm.
Response 13: We agree and have made some corrections: MSCs were cultivated in a medium without xenogenic components - MesenCult (Stem Cell, Canada) with the addition of 100 U / ml of penicillin 100 U / ml of streptomycin; 2 mM glutamine from the 3rd to the 5th passage.
Point 14: line112: PQ?
Response 14: We agree and have made some corrections: All mistakes was fixed. The CMPL was placed in a tangential flow filtration system and concentrated 8.08 times with an inlet pressure of 40-52 psi and an outlet pressure of 8-12 psi.
Reviewer 2 Report
This is a descriptive study which demonstrates the wound healing effect of MSC derived from the placenta or secretome of the cells on damaged cause by radiation. The study is descriptive, and no mechanisms of action are provided. The secretome is not characterized. The novelty of the study is questionable as the use of MSCs for wound healing process has been previous covered in many studies. It si not clear what is the added value of this study over many studies discussing this therapeutic venue. However, the dynamic changes in the healing process as shown in Figure 4, is new, and the focus of the manuscript should be on this part of the study.
Specific comments
This study needs rigorous English style editing.
Introduction
The authors should include additional information about MSC in cancer and healing process, in order to stress the difference between their study and the studies previously published.
The role of MSC in various biological processes should be mentioned and indicated as part of the mechanism of therapeutic activity
Materials and Methods
The fact that the animals are rats should be mentioned.
Line 68 - remove the word “randomly”.
Group 3, 4 description is not clear. Are they irradiated mice? (line 72-75).
Group 4, this reviewer believes that CMPL is both CM and PL – it is not mentioned in the Materials and methods section.
There is no mention whether the animals and groups were blinded to the evaluators of the wound healing process.
Line 87 - time of exposure has not been mentioned.
Line 107 – it is not clear whether the cells are CD45 positive or negative and whether 7aad is positive or negative.
Results
Representative flow cytometry plot of MSC characterization should be provided in Supplemental online material.
Figure 1:
Author should indicate groups’ name for clarity.
It is not clear what is “changed skin”- is it healed skin?
What is sm2? This reviewer is not familiar with this term.
The graphs lack statistical analysis.
Figure 2:
It is advised to include staining of collagen or sirus red to see the healing process and the formation of scar.
Figure 3:
The staining using DAB or any other methods is not clear in the figure. It could be that the resolution of the figure is blurry. Another possibility is weak staining to detect the specific markers. Higher magnification should also be provided in Figure 2 and not only 3.
In vitro analysis of angiogenesis can be performed to validate the results seen.
A summary graph demonstrating the dynamic changes in tissue parameters over time for each marker as shown in Figure 3 would increase clarity.
Figure 4:
The difference between blue and orange has not been mentioned in the figure nor in the figure legend.
Conclusion
The fact that the authors see a “wave” mentioned as part of the healing process and the angiogenesis over time has not been discussed. What is the meaning of this wave?
In summary
A descriptive study with minimal novelty. The manuscript should be re-written in order to focus on the novel parts. It is advised to demonstrate some mechanisms of action, better characterization of conditioned medium, and in vitro assays to show the healing process mentioned.
Author Response
Response to Reviewer 2 Comments
Dear REVIEWER,
Thank you for your work on the article. We revised the manuscript corrections are highlighted and figures in the attached files.
With best regards,
Authors.
And we clarify following questions:
Point 1: Comments and Suggestions for Authors
This is a descriptive study which demonstrates the wound healing effect of MSC derived from the placenta or secretome of the cells on damaged cause by radiation. The study is descriptive, and no mechanisms of action are provided. The secretome is not characterized. The novelty of the study is questionable as the use of MSCs for wound healing process has been previous covered in many studies. It si not clear what is the added value of this study over many studies discussing this therapeutic venue. However, the dynamic changes in the healing process as shown in Figure 4, is new, and the focus of the manuscript should be on this part of the study.
Response 1: We agree and have made some corrections. The innovation of this study lies in the use of the placenta as a source of mesenchymal stem cells.
Point 2: Specific comments
This study needs rigorous English style editing.
Response2: We agree and have made some corrections. English has been fixed.
Point 3: Introduction
The authors should include additional information about MSC in cancer and healing process, in order to stress the difference between their study and the studies previously published.
Response 3: The aim of our study was to find out whether the use of placental MSCs in the therapy of MSCs has prospects, but was not to study the processes of oncogenicity upon the introduction of MSCs and their products in LRI therapy.
Point 4: The role of MSC in various biological processes should be mentioned and indicated as part of the mechanism of therapeutic activity
Response 4: We agree and have made some corrections. Data supplemented. MSCs are capable of self-renewal and various types of differentiation in the adipogenic, osteogenic, chondrogenic, and myogenic directions. The use of MSCs leads to the healing of the wound surface of the skin and its appendages, diabetic ulcers, damage to skeletal muscles and cartilage, and the heart. Intravenous, local administration of MSCs helps to reduce necrotic changes, reduce inflammation, significantly improve the processes of granulation, reepithelialization, neoangiogenesis and hair restoration.
Paracrine factors initiate the stimulation of host MSCs, triggering the regeneration of damaged tissues. Thus the cytokines involved in the regulation of the inflammatory process include IL-1β, 4, 6, 10, 12, 17, TNF-α, TGF-β1, PGE-2, PDGF, HGF, SDF-1, etc .; VEGF, FGF-2, EGF, TGF-α, HGF, IGF-1, etc. , are responsible for neoangiogenesis, and the regulation of tissue fibrosis with the participation of IL-4, 16, TGF-β1, HGF, bFGF, etc. MSC secretome injections in the form of conditioned media containing extracellular vesicles also have positive effects, as well as the use of the MSCs themselves. Further assessment of the paracrine potential of MSCs may open up new ways of treating acute and chronic forms of MLP of the skin.
Sources of MSCs are various human tissues (bone marrow, adipose tissue, skin, placenta, synovial membrane, cartilage, etc.).
Point 5: Materials and Methods
The fact that the animals are rats should be mentioned.
Response 5: We indicated in the materials and methods the information on MSCs and SCs that these are rats.
Point 6: Line 68 - remove the word “randomly”.
Response 6: We agree and have made some corrections. All mistakes was fixed. Laboratory animals were randomized and divided into 4 groups (20 animals each) depending on the type of therapy performed.
Point 7: Group 3, 4 description is not clear. Are they irradiated mice? (line 72-75).
Response 7: We indicated that the rats were irradiated
Point 8: Group 4, this reviewer believes that CMPL is both CM and PL – it is not mentioned in the Materials and methods section.
Response 8: CMPL is not error, meaning the group CMPL is not a group that has had 2 treatments at the same time. In group 4, we used a conditioned medium concentrate, which was obtained as a result of culturing placental MSCs.
Point 9: There is no mention whether the animals and groups were blinded to the evaluators of the wound healing process.
Response 9: We agree and have made some corrections. We did not randomize blindly.
Point 10: Line 87 - time of exposure has not been mentioned.
Response 10: We added data to the section " Modeling of LRI ":
Modeling of relatively “soft” X-ray radiation of LRI was carried out on an LNK-268 (RAP100-10) X-ray unit («Diagnostika-M» LLC, Russia) with a radiation exposure mode at a dose of 110 Gy with an aluminum filter 0.1 mm, voltage 30 kV, beam current 6.1 mA, dose rate 21.1 Gy / min for 312 seconds (dose accuracy ± 5%, dose measurement uncertainty ± 6%) according to the proposed earlier method [10], leading to a short latency period and chronic skin ulcer in laboratory animals. After irradiation, the animals were seated in individual sterile boxes with an autonomous Smart Flow ventilation system (Tecniplast Group, Italy), providing free access to water and food.
Point 11: Line 107 – it is not clear whether the cells are CD45 positive or negative and whether 7aad is positive or negative.
Response 11: Data corrected: CD45- and 7AAD.
Point 12: Results
Representative flow cytometry plot of MSC characterization should be provided in Supplemental online material.
Response 12: There are many pictures in our article. Especially for you we can send a file to review our results. However, we decided not to insert this data so as not to overload the article.
Figure 1. Placental MSC immunophenotype: CD90 + / CD105 + / CD45− / CD34− / HLA-DR−, 7 − ADD (99.5%)
Point 13: Figure 1:
Author should indicate groups’ name for clarity.
Response 13: We have completed the transcript for our groups:
Figure: 1.a) Dynamics of development of visible region rats skin changes which was occurred under X-ray radiation, b) Dynamics of development of an open wound surface of the rats skin after exposure to X-ray radiation, c) Dynamics of healing of an open wound surface of the skin in animals (C - irradiated rats without subsequent therapy, CM - irradiated rats was received intradermal administration of culture medium (MesenCult) concentrate, PL - therapy of LRI using MSC of placenta, CMPL – therapy of LRI using a concentrate of conditioned medium collected from culture of the placental MSC).
Point 14: It is not clear what is “changed skin”- is it healed skin?
Response 14: It’s mean that zone of changes on the surface of the rat's body that we see with the naked eye. This includes the areas of hair loss, discoloration , of an unhealed (not epithelized) wound defect, and so on. In the phrase "wound surface" we mean only the unhealed area of the LRI.
Figure 2. Dynamics of the LRI of animals: a) Dynamics of development of visible region rats skin changes which was occurred under X-ray radiation, b) Dynamics of development of an open wound surface of the rats skin after exposure to X-ray radiation, c) Dynamics of healing of an open wound surface of the skin in animals (C - irradiated rats without subsequent therapy, CM - irradiated rats was received intradermal administration of culture medium (MesenCult) concentrate, PL - therapy of LRI using MSC of placenta, CMPL – therapy of LRI using a concentrate of conditioned medium collected from culture of the placental MSC)
Point 15: What is sm2? This reviewer is not familiar with this term.
Response 15: It’s our mistake, “sm2,” meant “cm2.”. Fixed the error.
Point 16: The graphs lack statistical analysis.
Response 16: We decided to indicate the reliability of the differences in Table 1 and Table 2, so as not to overload images with graphs.
Table signature:
Notes:
1 - significant differences in the group compared to the control (C) (p≤0.05),
2 - significant differences between the PL or CMPL group compared with the CM group (p≤0.05),
3 - significant differences between the PL and CMPL groups (p≤0.05).
Point 17: Figure 2:
It is advised to include staining of collagen or sirus red to see the healing process and the formation of scar.
Response 17: To obtain histological results samples were stained with Hematoxylin-Eosin. We cannot detect collagen in these samples. Perhaps we will do this with a different staining kit and provide this data in our next articles.
Point 18: Figure 3:
The staining using DAB or any other methods is not clear in the figure. It could be that the resolution of the figure is blurry. Another possibility is weak staining to detect the specific markers. Higher magnification should also be provided in Figure 2 and not only 3.
Response 18: We will provide these figures in good quality in a separate file.
Point 19: In vitro analysis of angiogenesis can be performed to validate the results seen.
Response 19: We have not done this research. We plan to conduct such studies in the future.
Point 20: A summary graph demonstrating the dynamic changes in tissue parameters over time for each marker as shown in Figure 3 would increase clarity.
Response 20: We will provide these figures in good quality in a separate file.
Point 21: Figure 4: The difference between blue and orange has not been mentioned in the figure nor in the figure legend.
Response 21: We agree and have made some corrections.
Figure 4. Immunohistochemical study of animal skin defects: a) The absolute number of IHC markers in samples of excised tissues of a skin defect of animal skin per 10 fields of view; b) Semi-quantitative assessment of the expression of IHC markers in excised samples of animal skin defect (C - irradiated rats without subsequent therapy, CM - Irradiated rats was received intradermal administration of culture medium (MesenCult) concentrate, PL - therapy of LRI using MSC of placenta, CMPL – therapy of LRI using a concentrate of conditioned medium collected from culture of the placental MSC. Blue - 28 days; orange - 112 days from the start of therapy. In score from 0 to 3, where 0 is the absence of expression, and 3 is the expressed expression).
Point 22: Conclusion
The fact that the authors see a “wave” mentioned as part of the healing process and the angiogenesis over time has not been discussed. What is the meaning of this wave?
In summary
Response 22: This issue needs to be discussed. There is still no clear answer, so our research is still ongoing.
Point 23: A descriptive study with minimal novelty. The manuscript should be re-written in order to focus on the novel parts. It is advised to demonstrate some mechanisms of action, better characterization of conditioned medium, and in vitro assays to show the healing process mentioned.
Response 23: We have characterized our medium in the section "Obtain an conditioned medium".
Reviewer 3 Report
In this research, the author confirmed that MSCs and their conditioned medium can repair X-ray damaged skin through detection of area of skin healing, histological examination and immunohistochemical study. The idea of applying MSCs derived from placenta to treat local radiation injuries of skin is new. However, there is too little experiment content to support the regeneration and epithelization of damaged skin after applying MSCs and their conditioned medium. Meanwhile, insufficient digging of related mechanisms makes the article a bit shallow.
Abstract:
1.“local radiation injuries (LRS)” (line 14) should be changed to“local radiation injuries (LRI)”.
2.Grammatical errors:“There were used……”(line 16) should be replaced with “We used……”;“was 6.7 times smaller than”(line 21) has semantic problems.
Introduction: -
Material and methods:
1.The details of groups of experiments can be replaced with a schematic diagram shown in the results rather than in the material and methods.
2.Y-axis in figure1a and 1b should be more specific.
Results and figures:
1.The results of immunological characteristics and viability of MSCs should be shown in the figure.
2.The subtitle “3.2.Planimetric analysis” is not very clear to understand.
3.What is “sm2”?
4.The subtitles“3.3. Histological examination” and“3.4.Immunohistochemical study” can be replaced with more specific results.
5.Too much description in “3.3” and too little annotation for figure 2.
6.The figures of immunohistochemistry alone cannot effectively explain the changes of various proteins, and more experiments such as western blot or PCR are needed to verify the results.
Discussion and conclusion:
1.The significance of the results for future research does not shown.
Author Response
Response to Reviewer 3 Comments
Dear REVIEWER,
Thank you for your work on the article. We revised the manuscript corrections are highlighted and figures in the attached files.
With best regards,
Authors.
And we clarify following questions:
Point 1: Comments and Suggestions for Authors
In this research, the author confirmed that MSCs and their conditioned medium can repair X-ray damaged skin through detection of area of skin healing, histological examination and immunohistochemical study. The idea of applying MSCs derived from placenta to treat local radiation injuries of skin is new. However, there is too little experiment content to support the regeneration and epithelization of damaged skin after applying MSCs and their conditioned medium. Meanwhile, insufficient digging of related mechanisms makes the article a bit shallow.
Response 1: We agree and have made some corrections.The data was supplemented.
Point 2: Abstract:
1.“local radiation injuries (LRS)” (line 14) should be changed to“local radiation injuries (LRI)”.
Response 2: We agree and have made some corrections. Data Fixed.
Point 3: 2.Grammatical errors:“There were used……”(line 16) should be replaced with “We used……”;“was 6.7 times smaller than”(line 21) has semantic problems.
Introduction: -
Response 3: We agree and have made some corrections. Data Fixed.
Point 4: Material and methods:
1.The details of groups of experiments can be replaced with a schematic diagram shown in the results rather than in the material and methods.
Response 4: Thanks for the suggestion. We decided to keep our text.
Laboratory animals were randomized and divided into 4 groups (20 animals each) depending on the type of therapy performed:
Group 1 - control (C)- irradiated rats without subsequent therapy;
Group 2 – control (CM) - Irradiated rats was received intradermal administration of culture medium (MesenCult) concentrate around the affected area three times on days 1, 14 and 21;
Group 3 - irradiated rats was received intradermal administration MSCs of human placenta (PL) three times on days 1, 14, and 21;
Group 4 - - irradiated rats was received intradermal administration of concentrate of conditioned medium (CMPL) MSCs of human placenta three times on days 1, 14, and 21.
Point 5: 2. Y-axis in figure1a and 1b should be more specific.
Response 5: We agree and have made some corrections.
Figure: 2. Dynamics of the LRI of animals: a) Dynamics of development of visible region rats skin changes which was occurred under X-ray radiation, b) Dynamics of development of an open wound surface of the rats skin after exposure to X-ray radiation, c) Dynamics of healing of an open wound surface of the skin in animals (C - irradiated rats without subsequent therapy, CM - irradiated rats was received intradermal administration of culture medium (MesenCult) concentrate, PL - therapy of LRI using MSC of placenta, CMPL – therapy of LRI using a concentrate of conditioned medium collected from culture of the placental MSC).
Point 6: Results and figures:
1.The results of immunological characteristics and viability of MSCs should be shown in the figure.
Response 6: There are many pictures in our article. Especially for you we can send a file to review our results. However, we decided not to insert this data so as not to overload the article.
Figure 1. Placental MSC immunophenotype: CD90 + / CD105 + / CD45− / CD34− / HLA-DR−, 7 − ADD (99.5%)
Point 7: 2.The subtitle “3.2.Planimetric analysis” is not very clear to understand.
Response 7: Planimetric analysis involves the measurement and assessment of wound defects. We can write it as "Visual assessment and measurement of LRI".
Point 8: 3.What is “sm2”?
Response 8: It’s our mistake, “sm2” meant “cm2”. Fixed the error.
Point 9: 4.The subtitles“3.3. Histological examination” and“3.4.Immunohistochemical study” can be replaced with more specific results.
Response 9: Thank you for your comment. We decided to show the detailed pathogenesis within the lesion.
Point 10: 5.Too much description in “3.3” and too little annotation for figure 2.
Response 10: Added a description to the picture:
(C - irradiated rats without subsequent therapy, CM - irradiated rats was received intradermal administration of culture medium (MesenCult) concentrate, PL - therapy of LRI using MSC of placenta, CMPL – therapy of LRI using a concentrate of conditioned medium collected from culture of the placental MSC).
Point 11: 6.The figures of immunohistochemistry alone cannot effectively explain the changes of various proteins, and more experiments such as western blot or PCR are needed to verify the results.
Response 11: Western blotting or PCR was not performed in this study. In the future, we plan to conduct these studies
Point 12: Discussion and conclusion:
1.The significance of the results for future research does not shown.
Response 12: Further research is needed to understand the mechanisms of action of placental MSCs in LRI of the skin.
Round 2
Reviewer 1 Report
--
Author Response
Dear REVIEWER,
Thank you for your work on the article. We revised the manuscript corrections using MDPI provides an English editing service.
With best regards,
Authors.

Reviewer 2 Report
The authors addressed my comments
Yet - novelty - there is a company that takes MSC from placenta - so it is not new.
Author Response
Dear REVIEWER,
Thank you for your work on the article. We revised the manuscript corrections using MDPI provides an English editing service.
With best regards,
Authors.
And we clarify following questions:
Point 1: Yet - novelty - there is a company that takes MSC from placenta - so it is not new.
Response 1: The innovation of our study lies in the fact that MSCs from the placenta are used in MLI therapy.

Reviewer 3 Report
- There is still too little research on related deep mechanisms.
- “local radiation injuries (LRS)” in the abstract is still wrong.
- The content of the subtitles (3.2;3.3;3.4) is not specific enough.
- The wrong spelling of “sm2” remains unchanged in table 1 and 2.
Author Response
Dear REVIEWER,
Thank you for your work on the article. We revised the manuscript corrections using MDPI provides an English editing service.
With best regards,
Authors.
And we clarify following questions:
Point 1: There is still too little research on related deep mechanisms.
Response 1: We are continuing our research to understand these mechanisms. In the future, our publications will be devoted to this.
Point 2: “local radiation injuries (LRS)” in the abstract is still wrong.
Response 2: We agree and have made some corrections (on "LRI").
Point 3: The content of the subtitles (3.2;3.3;3.4) is not specific enough.
Response 3: We have presented the content of the subtitles (3.2; 3.3; 3.4) in this version when writing the article.
Point 4: The wrong spelling of “sm2” remains unchanged in table 1 and 2.
Response 4: We agree and have made some corrections in Tables 1 and 2 on cm2.
